# The Effect of *Laminaria japonica* on Metabolic Syndrome: A Systematic Review of Its Efficacy and Mechanism of Action

**DOI:** 10.3390/nu14153046

**Published:** 2022-07-25

**Authors:** In-Seon Lee, Seok-Jae Ko, Yu Na Lee, Gahyun Lee, Md. Hasanur Rahman, Bonglee Kim

**Affiliations:** 1Department of Meridians and Acupoints, College of Korean Medicine, Kyung Hee University, Seoul 05253, Korea; inseon.lee@khu.ac.kr; 2Acupuncture & Meridian Science Research Center, Kyung Hee University, Seoul 02447, Korea; 3Department of Gastroenterology, College of Korean Medicine, Kyung Hee University, Seoul 05253, Korea; kokoko119@khu.ac.kr; 4College of Korean Medicine, Kyung Hee University, Seoul 05253, Korea; unaly@khu.ac.kr (Y.N.L.); dlrkgusl77@khu.ac.kr (G.L.); hasanurrahman.bge@gmail.com (M.H.R.)

**Keywords:** metabolic syndrome, *Laminaria japonica*, atherosclerosis, diabetes, obesity, AMPK

## Abstract

Metabolic syndrome (MetS) is a medical condition characterized by abdominal obesity, insulin resistance, high blood pressure, and hyperlipidemia. An increase in the incidence of MetS provokes an escalation in health care costs and a downturn in quality of life. However, there is currently no cure for MetS, and the absence of immediate treatment for MetS has prompted the development of novel therapies. In accordance with recent studies, the brown seaweed *Laminaria japonica* (LJP) has anti-inflammatory and antioxidant properties, and so forth. LJP contains bioactive compounds used as food globally, and it has been used as a medicine in East Asian countries. We conducted a systematic review to examine whether LJP could potentially be a useful therapeutic drug for MetS. The following databases were searched from initiation to September 2021: PubMed, Web of Science, EMBASE, and Cochrane Central Register of Controlled Trials Library. Clinical trials and in vivo studies evaluating the effects of LJP on MetS were included. LJP reduces the oxidative stress-related lipid mechanisms, inflammatory cytokines and macrophage-related chemokines, muscle cell proliferation, and migration. Bioactive-glucosidase inhibitors reduce diabetic complications, a therapeutic target in obesity and type 2 diabetes. In obesity, LJP increases AMP-activated protein kinase and decreases acetyl-CoA carboxylase. Based on our findings, we suggest that LJP could treat MetS, as it has pharmacological effects on MetS.

## 1. Introduction

Metabolic syndrome (MetS) is characterized by abdominal obesity, insulin resistance, hypertension, and hyperlipidemia [1]. Cancer, diabetes, atherosclerosis, obesity, and hyperlipidemia disrupt the metabolic balance and are associated with MetS, which is caused by impediments to anabolism and catabolism in the human body [2]. Hyperglycemia, hyperlipidemia, hyperinsulinemia, hypertension, and atherosclerosis are symptoms of diabetes mellitus (DM), a chronic metabolic condition that affects the body’s ability to process carbohydrates, lipids, and lipoproteins [3]. The increased consumption of high-calorie and low-fiber food and decreased physical activity result in a higher incidence of MetS, leading to high health care costs and poor quality of life [1]. In recent decades, the prevalence of MetS has reached an epidemic-level, with approximately 2 billion adults affected and 1/3 of children being obese; this is a significant public health and clinical concern worldwide [4,5]. However, there is currently no cure for MetS, but certain disciplined lifestyles and prescribed drugs could prevent the long-term effects of the disease. A broad range of therapies has been approved by the United States Food and Drug Administration (FDA) that address the different biological systems and mechanisms implicated in this disease. There are 59 FDA-approved antihyperglycemic medicines that have been unique since the FDA approved human insulin in 1982. A total of 36 novel molecular entities (NMEs) as monotherapies and 23 unique pharmacological combinations of two or more antihyperglycemic medicines have also been licensed. NMEs, recently licensed as monotherapies, are based on molecular pathways that have been validated by other authorized antihyperglycemic medicines [6]. There is no guarantee of full recovery by the FDA-approved drugs for MetS, and they also have some long-term and short-term side effects including flatulence, abdominal bloating and discomfort, diarrhea, and an increase in bladder cancer [7]. Furthermore, non-selective drug resistance and bioavailability are some of the drawbacks of the conventional therapies for MetS [1]. Therefore, developing novel drugs for the treatment of MetS is urgently required.

Brown seaweed is an excellent source of carbohydrates, vital amino acids, vitamins, minerals, and other nutrients. According to recent studies, brown seaweed contains various phenolic compounds with anti-inflammatory, antioxidant, and anticarcinogenic properties [8]. Laminaria species are a vital marine medicinal food source that have biological effects on the human body [9], wherein *Laminaria japonica* (LJP) is a rich source of algae, minerals, and dietary fiber as well as a vital source of dietary fiber production [10]. In Korea, Japan, and China, LJP is cultured as one of the most economically important seaweeds [11]. It is a traditional food and medicine in Asia due to its pharmacological and biochemical qualities. It is also known as kelp, sea tangle, kombu or kunbu in Japan, and dashima in Korea [1].

According to recent studies, polysaccharides are the most crucial physiologically active components of LJP. Algin, laminarin, fucoidan, varying quantities of galactose, xylose, and glucuronic acid as well as a small amount of protein are the main components of the natural macromolecules found in the intercellular or intracellular layer of the LJP [3]. LJP has multiple pharmacological effects including anti-tumor, anti-thrombotic, anti-atherosclerosis, hypolipidemic, hypoglycemic, antioxidant, anti-inflammatory, renoprotective, and immunomodulatory [12]. In addition, LJP protects against several metabolic syndromes including diabetes, obesity, and atherosclerosis [3,13,14]. Some in vitro studies have also identified an antithrombotic effect in LJP [15]. However, there is no systematic review of studies testing for the effects of LJP on MetS, and we aimed to address the role of LJP on the treatment of MetS.

In this study, we conducted a systematic review to determine the effects and mechanisms of LJP in treating patients with MetS and the possibility of using LJP as a therapeutic drug. We also discuss the mechanisms to delve deeper into its molecular pharmacology and highlight the recent progression of its therapeutic developments. This review is expected to direct future research toward a better understanding of how LJP could modulate MetS and be developed as a potential therapeutic component for the treatment of MetS.

## 2. Materials and Methods

### 2.1. Search Strategy and Registration

A systematic review was conducted to retrieve and critically appraise the available evidence on the effects of LJP on MetS. The literature search was performed in September 2021 using four databases: PubMed, Web of Science, EMBASE, and the Cochrane Central Register of Controlled Trials Library. We followed the Preferred Reporting Items for Systematic Reviews and Meta-Analyses (PRISMA) statement to conduct the present systematic review [16]. Intervention-related terms such as ‘LJP’, ‘kelp’, ‘kombu’, ‘kunbu’, and disease-related terms ‘metabolic’, ‘cardiovascular’, ‘hypertension’, ‘glucose’, ‘diabetes’, ‘insulin resistance’, ‘cholesterol’, ‘obese’, etc., were used in the search. The search strategy for each database can be found in the Appendix A. This systematic review was registered in the Open Science Framework registry (https://osf.io/g2bqk, accessed on 5 September 2021) and the protocol has been previously published [1].

### 2.2. Eligibility Criteria

The eligibility criteria were based on the population (P), intervention (I), comparison (C), outcome (O), and study design (S) (PICOS) strategy. Clinical trials and in vivo studies that evaluated the effect of LJP on MetS including obesity, diabetes, hyperlipidemia, fatty liver, and atherosclerosis were included. We excluded duplicate articles, those that were not written in English, original studies, non-LJP-related studies, in vitro studies, or disease-related studies.

### 2.3. Data Extraction

The process of data extraction from the studies in this systematic review are summarized in Figure 1, providing the following information: author, year of publication, sample size, characterization of the study population, interventions, controls (if applicable), clinical outcomes, results, and adverse events (in clinical trials only).

### 2.4. Quality Assessment and Risk of Bias

To ensure that the results of the evidence synthesis are transparent, the risk of bias assessment was performed for randomized controlled trials (RCT) using the Revised Cochrane risk-of-bias tool for randomized trials (RoB 2) for individually randomized RCTs [17]. RoB 2 for cross-over trials was used in the cross-over RCT study. Following the guidelines, we assessed each study’s risk of bias based on the randomization process, the deviations from the intended interventions, missing outcome data, outcome measurement, the selection of the reported result, and period and carry-over effects (only for the cross-over study). Cochrane Methods (https://methods.cochrane.org/bias/resources/rob-2-revised-cochrane-risk-bias-tool-randomized-trials, accessed on 24 September 2021) offers an Excel tool that we used to calculate the risk of bias and to generate a table of the assessment result.

### 2.5. Literature Search

Of the 727 studies found in the search (PubMed, *n* = 145; Web of Science, *n* = 396; EMBASE, *n* = 168; and Cochrane Central Register of Controlled Trials Library, *n* = 18), the titles and abstracts of 457 articles and the full texts of 73 articles were screened after removing any duplicates (*n* = 270). A total of 419 articles were excluded (not written in English, *n* = 9; not an original study, *n* = 39; not related to LJP, *n* = 133; an in vitro study, *n* = 28; and not related to MetS, *n* = 212). Finally, 36 articles (eight clinical trials and 28 in vivo studies) were included in the present systematic review (Figure 1).

### 2.6. Studies Characteristics

Of the 36 studies, eight evaluated the effect of LJP on MetS in humans. Of the remaining 28 animal studies, 10 evaluated the effect of LJP on diabetes, seven on obesity, eight on atherosclerosis, and three on hyperlipidemia/fatty liver. All clinical trials were conducted on healthy participants.

## 3. Results

### 3.1. Animal Studies

#### 3.1.1. Diabetes

The anti-diabetic effects of LJP have been reported in 10 studies (Table 1). Jia et al. [3] reported that polysaccharides from LJP had hypoglycemic and hypolipidemic effects in mice with alloxan-induced diabetes. Twenty-eight days of LJP administration prevented body weight loss, decreased the glucose levels, and increased the insulin levels. It also ameliorated the alloxan-induced alterations in lipid metabolism.

Li et al. [18] demonstrated the hypoglycemic effect of LJP in a mouse model of type 2 DM. They suggested that LJP may partially recover the secretory function of islet cells, leading to improved regulation of glucose metabolism. Two weeks of oral LJP administration significantly reduced the fasting glucose levels and increased serum insulin and amylin levels in alloxan-induced DM mice.

Bu et al. [19] focused on the hypoglycemic effect of butyl-isobutyl-phthalate (BIP) as an active compound in the rhizoids of LJP, as determined by spectral analysis. BIP exhibited significant concentration-dependent inhibitory activity against α-glucosidase in vitro. In vivo, three days of oral administration of BIP significantly reduced the blood glucose levels in streptozotocin (STZ)-induced diabetic mice.

Jin et al. [20] reported that pretreatment with 100 mg/kg aqueous LJP extract for 5 days reduced glucose levels, hepatic lipid peroxidation, and xanthine activity in diabetic rats. The LJP extract may scavenge the reactive oxygen species or enhance the utilization efficiency of glutathione, thereby reducing the lipid peroxidation levels in the liver of diabetic rats.

Liang et al. [21] demonstrated the protective effect of low-molecular-weight fucoidan (LMWF) from LJP on lipid profiles and basal blood pressure by ameliorating oxidative stress, prostanoid production, and hyper-responsiveness of aortic smooth muscles in STZ-induced rats with type 1 DM.

Park et al. [22] found that 13 weeks of oral administration of sea tangle (LJP) powder (15% w/w) or sea tangle water extract (4% w/w) decreased the blood glucose levels and improved lipid metabolism in STZ-induced diabetic rats. This effect might be exerted by increased serum insulin levels and fecal excretion of lipids, preventing lipid absorption.

Xu et al. [23] investigated the effect of sulfated polysaccharides (SPS) extracted from LJP to relieve diabetic nephropathy (DN) in an STZ-induced diabetic model. The administration of gavage-fed SPS (200 mg/kg) for 80 days decreased the blood urea nitrogen, serum creatinine levels, and symptoms of DN. Changes in the expression of protein kinase C (PKC)α, PKCβ, P-selectin, nuclear factor kappa B (NFκ), and p65 showed that SPS regulates DN via the PKC/NF-κB pathway.

Long et al. [24] reported on the hypoglycemic effect of kelp (LJP) in alloxan-induced DM model rats. Two weeks of oral administration of kelp lowered the fasting blood glucose (FBG), malondialdehyde (MDA), and nitric oxide (NO). The shape and structure of the islet cells improved with the upregulation of superoxide dismutase (SOD) and the downregulation of inducible nitric oxide synthase (iNOS) in the kelp-treated group. Kelp might help recover the secreting function of islet cells and decrease the level of FBG through its antioxidant effect.

Wang et al. [25] investigated the anti-inflammatory, anti-thrombosis, and revascularization effects of LMWF extracted from LJP in diabetic peripheral arterial disease rats. LMWF exhibited a therapeutic effect on hindlimb ischemia in diabetic rats, likely by ameliorating the dysfunction of endothelial NOS and enhancing revascularization.

Kang et al. [8] investigated the antidiabetic effect and possible mechanism of LJP using in vitro and high-fed diet in vivo models. Sixteen weeks of oral administration of LJP induced insulin-signaling-related proteins such as the phosphorylation of protein kinase B in the skeletal muscle. LJP also affects glucose homeostasis by inhibiting α-glucosidase activity, increasing muscle glucose uptake, and regulating inflammatory cytokines in the skeletal muscle cells (Figure 2).

#### 3.1.2. Obesity

Seven studies reported the anti-obesity effects of LJP (Table 2). Oh et al. [13] reported that LJP had anti-obesity and anti-inflammatory effects in high-fat diet (HFD)-fed mice. The treatment was continued for 16 weeks, and the LJP was supplemented with 5% of the diet. LJP reduces blood glucose, leptin, and circulating cytokine levels.

Shang et al. reported that fucoidan in LJP had a beneficial effect on diet-induced MetS in HFD-fed mice [26]. In the 16-week experiment, 200 mg/kg of LJP was orally administered to mice. Fucoidan reduced the levels of total cholesterol (TC), triacylglycerides (TAG), and fasting blood glucose.

Jang and Choung reported that the ethanol extract of LJP had an anti-obesity effect in HFD-fed mice [27]. According to the study, the extract reduced the levels of serum triglyceride (TG), TC, and low-density lipoprotein cholesterol (LDL-C). It increased the high-density lipoprotein cholesterol (HDL-C), HDL-C/TC ratio, and adiponectin. It also increased the ratios of p-AMPK/AMPK and p-ACC/ACC.

Zhang et al. reported that the insoluble dietary fiber of LJP improved the obesity-related features in HFD-fed mice and played a role in gut microbiota dysbiosis and protection against HFD-induced liver injury [28]. It was shown that fiber reduced the levels of serum glucose, TC, alanine aminotransferase (ALT), and aspartate aminotransferase (AST) and restored the levels of acetate, propionate, and cecal short-chain fatty acid (SCFA).

Li et al. reported that polysaccharides from LJP could combat obesity by increasing p-AMPK and lowering the fatty acid synthase (FAS) and tumor necrosis factor (TNF)-α levels [29]. According to the experiment, high-fat mice administered with LJP polysaccharides experienced hypoglycemic effects, improved serum lipid profiles, and ameliorated intestinal damage.

Duan et al. reported that polysaccharides from LJP had an anti-obesity effect and normalized gut microbiota in HFD-fed mice [30]. In the experiment, 0.25% LJP solution was administered to mice as drinking water. LJP increases the HDL-C/LDL-C ratio and reduces the serum lipid levels.

Han et al. reported that LJP exerted a hypotriglyceridemic effect on HFD-fed mice by phosphorylating 5 AMP-activated protein kinase [31]. LJP was supplemented with 5% of the diet during the 16-week experiment. According to this study, LJP increased the fecal bile acid and reduced the FBG levels, plasma triglyceride levels, and hepatic lipid accumulation (Figure 3).

#### 3.1.3. Atherosclerosis

Eight studies elucidated the anti-atherosclerotic effects of LJP (Table 3). Most studies used mouse models with dietary interventions (high-fat, high cholesterol, and hyperlipidemia) to induce atherosclerosis. Yao et al. reported that a kelp extract obtained with citric acid instead of water significantly lowered the possibility of atherosclerosis by ABTS (2,2′-Azino-bis-(3-ethylbenzothiazoline-6-sulfonic acid) radical scavenging activity [14]. According to the experiment, high-fat mice administered the LJP citric acid extract experienced hypolipidemic and antioxidative effects and decreased the incidence of cardiovascular and cerebrovascular disease.

Li et al. reported that LJP polysaccharides can combat atherosclerosis by lowering various enzymes and receptors in the serum [32]. Furthermore, the study elucidated a causal relationship between LJP and enhanced insulin resistance by explaining the increased number of intestinal goblet cells and decreased Akkermansia in the serum of high-fat mice after ingesting LJP. In a mouse model in which a high-fat cholesterol diet produced atherosclerosis, Zha et al. examined the anti-atherosclerotic effect [38]. They discovered that 14 weeks of oral administration of polysaccharides derived from LJP (50 and 200 mg/kg/day) had significant hypolipidemic and anti-atherosclerotic benefits by modulating the hepatic insulin signaling system.

Peng et al. reported that LJP polysaccharides could combat atherosclerosis by increasing the number of SOD, thus inhibiting the phosphorylation of various enzymes and decreasing the amount of TNF [33]. According to this experiment, high-fat mice could undergo anti-atherosclerotic, hypolipidemic, and antioxidative effects.

Huang et al. reported that LJP polysaccharides could combat atherosclerosis by increasing the number of lipoprotein lipases and accelerating lecithin cholesterol acyltransferase activity [34]. According to the experiment, high-fat mice experienced hypoglycemic and anti-atherosclerotic effects of cardiovascular disease after ingesting LJP for 8 weeks.

Zheng et al. reported that LJP polysaccharides could combat atherosclerosis by increasing the number of G protein-coupled receptors and carnitine palmitoyltransferase-1A (CPT-1A) [35]. Simultaneously, peroxisome proliferator-activated receptor-γ (PPAR-γ) and TNF-α were downregulated, which ensured enhanced anti-obesity effects.

Zha et al. investigated the effects of LJP polysaccharides at a dose of 400 mg/kg/day [36]. The total serum TC, TG, high-density lipoprotein-cholesterol (HDL-C), and low-density lipoprotein-cholesterol (LDL-C) levels were reduced, while the antioxidant enzyme activities were increased.

Wang et al. studied the anti-atherosclerotic, hypolipidemic, and anti-inflammatory effects of fucoidan derived from LJP in a mouse model of high-cholesterol diet-induced atherosclerosis [37]. They discovered that either 50 mg/kg/day or 100 mg/kg/day of fucoidan for 16 weeks lowered the serum lipid levels (TG, TC, LDL-C, HDL-C), atherosclerotic plaque growth, lectin-like oxidized low-density lipoprotein receptor-1 (LOX-1) expression, and pro-inflammatory mediator expression (e.g., interleukin-6, TNF-α, intercellular adhesion molecule 1 (ICAM-1), and vascular cell adhesion molecule 1 (VCAM-1) (Figure 4).

#### 3.1.4. Hyperlipidemia/Fatty Liver

Two studies of anti-hyperlipidemia and fatty liver effects of LJP were reposted (Table 4). Zhang et al. (2020) reported that macroalgae LJP (MLJ) was a potential treatment for hyperlipidemia as high-fat mice were able to undergo hypolipidemic effects after being administered with MLJ [39]. The mechanism underlying this efficacy was increased SOD and glutathione peroxidase (GSH-Px).

Zhang et al. reported that *Lactobacillus brevis* FZU0713-fermented LJP (FLJ) had a beneficial effect in hyperlipidemic rats [40]. According to a previous study, the oral administration of FLJ regulated hepatic mRNA levels of the gene, thereby reducing the serum levels of TC and TG and increasing the fecal levels of acid acetate, propionate, and isobutyrate.

A dieckol-enriched extract of LPJ was found in a HFD-induced non-alcoholic fatty liver disease mouse model, which had a substantial hypolipidemic impact [41]. They discovered that taking 5 mg/kg of polyphenols orally for four weeks decreased the visceral fat index, plasma and hepatic lipid levels (TG, TC, HDL, LDL), and hepatic steatosis.

### 3.2. Clinical Studies

Four clinical trials evaluated the effects of LJP on the body fat and fat-related parameters (Table 5). You et al. evaluated the effect of a body weight control program that included 20 g of LJP per day [42]. They discovered that consuming LJP lowered the body weight, fat mass, waist–hip ratio, and body mass index and improved the quality of life and physical functioning. The results should be interpreted with caution because the program incorporated exercise and nutrition education, and the decrease in blood cholesterol levels was insignificant. Nishimura et al. conducted a clinical study to support the effect of LJP on body fat, gastrointestinal symptoms, and blood measurements [43]. They found that dried young LJP powder improved gastrointestinal symptoms including decreased passage of stools. The lipid profile did not improve, although it lowered body fat and serum triglyceride levels (in those with excessively high serum triglyceride levels). Nishiumi et al. studied serum lipid and molecular profiles in participants with excessively high serum triglyceride levels before and after consuming LJP [44]. They found that roasted LJP ameliorated gastrointestinal symptoms; an example is the decreased passage of stools. It reduced the body fat and serum triglyceride levels only in patients with excessively high serum triglyceride levels. They reanalyzed the data [45] and found that phosphatidylcholines with diacyl links, lysophosphatidylcholine/diethanolamine with acyl linkages, and free fatty acids were improved after LJP use. Aoe et al. conducted a double-blind clinical trial employing iodine-reduced powder of LJP and found that it significantly reduced the body fat percentage without affecting thyroid function [46].

Kang et al. studied the antioxidant effects of fermented LJP in participants with high gamma-glutamyl transferase (GGT) levels [47]. Consumption of fermented LJP for four weeks decreased the serum levels of GGT and malondialdehyde while increasing the catalase (CAT) and SOD activities. These findings suggest that fermented LJP possesses antioxidant and hypolipidemic properties. The effect of LJP on gut microbiota in healthy participants was explored by Ko et al. [48]. However, no significant changes in the number of microbiomes, gastrointestinal symptoms, bowel function, or quality of life were observed. Only those who received LJP paired with probiotics (lactic acid bacteria) experienced significant alterations in their microbiome.

Choi et al. investigated the effects of FLJ on brain-derived neurotrophic factor-related muscle growth, lipolysis, and total lean mass in middle-aged female participants. They found that 8 weeks of LJP consumption significantly decreased the total fat mass and triglycerides in body composition as well as significantly increased the serum brain-derived neurotrophic factor (BDNF), angiotensin-converting enzyme, human growth hormone, insulin-like growth factor-1 levels, and total lean mass as well as improvements in the total work, knee extension, and flexion, implying that LJP may have anti-obesity properties and increase the release of muscle-related growth hormones [49].

### 3.3. Quality and Risk of Bias

The rating of most studies resulted in “some concerns”, and one cross-over RCT was found to have a “high risk” of bias, as the carryover effect was not controlled (no wash-out period). Most studies did not incorporate detailed information on the randomization, blinding, and allocation concealment; outcome assessor blinding was poorly reported (Table 6).

## 4. Discussion

MetS includes hypertension, abdominal obesity, hyperlipidemia, and insulin resistance [49]. Impaired metabolic homeostasis in the human body is the leading cause of this syndrome [2]. In this systematic review, we found that LJP could significantly improve the clinical outcomes in patients suffering from MetS and the laboratory measurements in in vivo models of MetS.

Several reviews have reported algae’s effects on lifestyle-related diseases including LJP. Shirosaki and Koyama introduced studies on the ability of LJP to prevent obesity and diabetes and some approaches for effective of the bioactivities found in LJP [9]. They showed that the potential usefulness of LJP as a functional food may be increased through different processing methods and the growth stage of LJP. They also conducted an in vitro study using Caco-2 cells and reported that LJP suppressed the transmission of glucose in monolayer membranes by inhibiting GLUT2, leading to decreased postprandial blood glucose levels. D’Orazio et al. showed that fucoxanthin (FX), a characteristic carotenoid present in brown seaweeds, and LJP had antidiabetic, anticancer, antioxidant, and anti-photoaging properties against lifestyle-related diseases [51]. Gabbia et al. demonstrated that various types of brown seaweeds could be used as a nutraceutical and functional food for treating MetS comorbidities [52]. The review suggested using seaweeds in young people and coeliac patients to supplement metabolic comorbidities. Popović-Djordjević et al. reported that bioactive compounds of algae such as brown algae directly enhance insulin secretion, prevent the formation of amyloid plaques, and decrease hyperglycemia in type 2 DM [53]. These reviews have several limitations including the lack of human clinical studies, focusing only on specific components of LJP, and targeting seaweeds other than LJP. There are various valid components of LJP (polysaccharides, phenolic compounds, or pigments), and the seaweed composition may vary according to algal species, degree of maturity, storage, and processing conditions after harvesting [54].

Atherosclerosis is a disease characterized by chronic inflammation of medium-and large-sized arteries and plaque formation in the intima. LJP inhibits atherothrombosis by (1) suppressing oxidative stress-related lipid mechanisms including the inhibition of LDL oxidation, the uptake of oxidized LDL, production of reactive oxygen species (ROS) and LOX-1, and enhancing the activities of antioxidant enzymes such as SOD and CAT; (2) reducing the secretion of inflammatory cytokines (e.g., IL-6, IL-1β, TNF-α, etc.) and macrophage-related chemokines; (3) interrupting the PI3K/Akt pathway; and (4) regulating the proliferation and migration of vascular smooth muscle cells. ROS plays an important role in LDL-C oxidation, which increases the expression of cell adhesion molecules including VCAM-1 and ICAM-1. They mediate the adhesion of leukocytes to the endothelial cells [55]. LOX-1 is one of the primary receptors of oxidized LDL, which facilitates oxidized LDL uptake and increases the expression of monocyte chemoattractant protein (MCP-1) and the apoptosis of endothelial cells. MCP-1 stimulates macrophage infiltration of the arterial wall during the early stages of atherogenesis [56,57]. The PI3K/Akt pathway thickens the arterial wall and induces plaque formation and vascular smooth muscle cell (VSMC) proliferation and contractility [58,59]. VSMC are a significant source of plaque cells; thus, they are implicated mechanistically at all stages of atherosclerosis [60]. Although more research is needed to understand the precise role of LJP in anti-atherosclerosis mechanisms, LJP may potentially treat atherosclerosis via antioxidant and anti-inflammatory mechanisms.

The mechanism of LJP in diabetes in this review can be summarized as follows: (1) the role of bioactive α-glucosidase inhibitors in ameliorating diabetic complications; (2) a therapeutic target in obesity and type 2 DM is lowering FFA; and (3) antioxidants reduce free radicals. α-Glucosidase, a carbohydrate-hydrolyzing enzyme, is generally competitively inhibited by α-glucosidase inhibitors, leading to delayed glucose absorption in the small intestine and ultimately controlling postprandial hyperglycemia [61]. This mechanism is a modern therapeutic approach for patients with type 2 DM (e.g., miglitol, acarbose, and voglibose are commercially available drugs). However, regular administration of these drugs results in various adverse events such as stomach pain and diarrhea [62]. Hence, LJP may be considered as a potent α-glucosidase inhibitor to reduce diabetic complications due to its immense efficacy in modulating postprandial hyperglycemia and minimal side effects. The relationship between glucose and FFA metabolism is well-described in the ‘Randle theory’, which demonstrates that FFA competes with glucose as an energy substrate in the muscle and adipose tissue; therefore, it inhibits the insulin signaling pathway, leading to insulin resistance in the skeletal muscle and liver [63]. Chronically elevated FFA levels also have a deleterious effect on the pancreas, which may increase the formation of NO, inducing β-cell apoptosis [64]. Natural compounds from LJP, polysaccharides, may be a therapeutic option for type 2 DM and obesity by reducing elevated FFA levels or insulin resistance. In addition, half of the studies in this review (5 out of 10) reported on the antioxidant defense mechanisms of LJP with oxidative stress biomarkers including GSH levels, GSH-Px, MDA, NO concentration, ROS, and SOD. Abnormally high levels of free radicals and subsequent activation of the transcription factor (NF-κB) have been linked to the enhancement of NO production, leading to islet β-cell damage and the development of late diabetic complications [65]. Interestingly, LJP lowered NO production (in islet cells of rats) and increased NO bioavailability (in endothelial cells of rat hindlimbs) to achieve the homeostasis of NO concentration [24,25]. Excessive NO production is involved in the development of pro-inflammatory reactions, tissue damage, and organ dysfunction [66,67]. Endothelium-derived NO is one of the most potent endogenous vasoprotective agents in normal vascular physiology because of its anti-inflammatory, anti-proliferative, and antithrombotic properties [68,69].

The anti-obesity mechanism of LJP in this review can be summarized as follows: (1) an increase in AMPK; (2) a decrease in ACC; and (3) a decrease in TNF-α and IL-6. Obesity is caused by a chronic imbalance in energy output and intake when excess energy accumulates in white adipose tissue instead of brown adipose tissue, which is difficult to prevent and cure in humans [70]. Notably, obesity is the leading cause of type 2 DM and insulin resistance [71]. AMPK is a key factor that regulates energy homeostasis and is used to target obesity-related drugs [72]. For example, *Rhus chinensis* Mill (Wu et al., 2021), *Chrysanthemum morifolium* flower [73], and sulforaphane in broccoli [74], etc. have shown anti-obesity effects through the regulation of AMPK. Since AMPK is a crucial molecule in obesity, most studies in this review showed an elevated expression of AMPK, decreasing p-SREBP1-c, p-ACC, CPT-1, and TNF-α. Fatty acids play an essential role in humans by acting as signaling molecules and structural components [75]. However, aberrant upregulation of fatty acid production can cause obesity. Inhibiting the core enzymes involved in fatty acid synthesis including ACC and FAS could be an attractive therapeutic target for obesity [76]. Inflammation is one of the critical processes associated with obesity, and the expression of TNF-α, an essential pro-inflammatory cytokine, is elevated in the adipose tissue of several experimental models of obesity [77,78]. LJP significantly decreased the expression of p-ACC and TNF-α in the studies reviewed.

The bioavailability of fucoidan and iodine extracted from LJP was also studied. Zhao et al. compared two types of fucoidan regarding their bioavailability, medium molecular weight (MMW) fucoidan, and LMWF in rats. They found that oral administration of LMWF resulted in better absorption and antithrombotic effects than MMW fucoidan. High-performance liquid chromatography (HPLC) analysis of fucoidan in the plasma and urine samples showed that 200, 400, and 800 mg/kg LMWF and 800 mg/kg MMW fucoidan significantly increased the plasma and urine levels of fucose, suggesting higher bioavailability of LMWF compared to that of MMW fucoidan [79]. Sun et al. compared various pretreatment methods for the bioavailability of iodine in LJP in vitro based on the ratio of absorbed and added iodine by/to cells. They found that boiled LJP had a higher bioavailability than steamed and soaked LJP. However, further studies are required [80]. In addition to fucoidan and iodine, alginates, polyphenols, and polysaccharides as bioactive nutritional compounds extracted from LJP have been evaluated as solutions for the treatment and prevention of Met-related diseases [81]. In this review, alginates, polyphenols and polysaccharides have been reported to possess anti-diabetic [3,10,19,24,33], anti-obesity [29,30,31,32,38], antioxidant [15,34,37], anti-inflammatory [37,38], and hypolipidemic [15,33,34,37] effect.

Although the interaction of LJP with other drugs or herbs has rarely been studied, fucoidan, a sulfated polysaccharide extracted from brown algae, has been studied in combination with other drugs, primarily chemotherapeutic drugs, in cancer patients. For example, in this study, fucoidan (extracted from *Cladosiphon okamuranus*) was administered with chemotherapeutic drugs (oxaliplatin plus 5-fluorouracil/leucovorin or irinotecan plus 5-fluorouracil/leucovorin). It showed potential anticancer effects against recurrent colorectal cancer by lowering the cytotoxic effects of chemotherapy drugs in patients with nausea, vomiting, stomatitis, diarrhea, and liver dysfunction [82]. In addition, fucoidan derived from *Undaria pinnatifida* was administered to breast cancer patients undergoing hormonal therapies using letrozole and tamoxifen, and the daily administration of 1 g of fucoidan for 3 weeks did not change the plasma levels of letrozole and tamoxifen or reports of toxicity [83].

The recent findings in this review support the idea of the effect of LJP on MetS. However, this study had several limitations. Due to the small number of studies and the diverse outcomes in the included studies, we could not perform a meta-analysis of the LJP clinical trials. Despite growing evidence from in vitro and in vivo studies, there is no substantial evidence to support the effect of LJP in humans. A detailed investigation of the mechanism is lacking including those of the individual studies in this review. For example, indices related to lipogenic and lipolytic metabolism, 3T3-L1 cells or PPAR-γ, and the activity of enzymes involved in hepatic glucose metabolism including glucose-6-phosphatase (G-6-Pase) and G-6-phosphate dehydrogenase (G-6-PDH) have not been investigated as yet. There is also a need to establish an adequate MetS model. Most studies on diabetes in this review (8 out of 10) used STZ or alloxan as the diabetogenic chemicals. STZ or alloxan destroys pancreatic beta cells to induce diabetes in an experimental model. Therefore, they may not well-represent type 2 DM [84]. Future studies on the toxicity of the long-term administration of LJP should be conducted.

In this review, we summarized the available evidence on the effects and mechanisms of LJP on MetS: obesity, diabetes, atherosclerosis, and hyperlipidemia (Figure 5). We found significant evidence of the antioxidant, hypoglycemic, and hypolipidemic effects of LJP in various animal models of MetS as well as in a few clinical trials. On the other hand, the safety, active components, extraction methods for better absorption, and the dose level of LJP should be tested both in vivo and in vitro. Based on this evidence, we suggest large clinical trials to demonstrate the effects of LJP in combination with conventional drugs.

## Figures and Tables

**Figure 1 nutrients-14-03046-f001:**
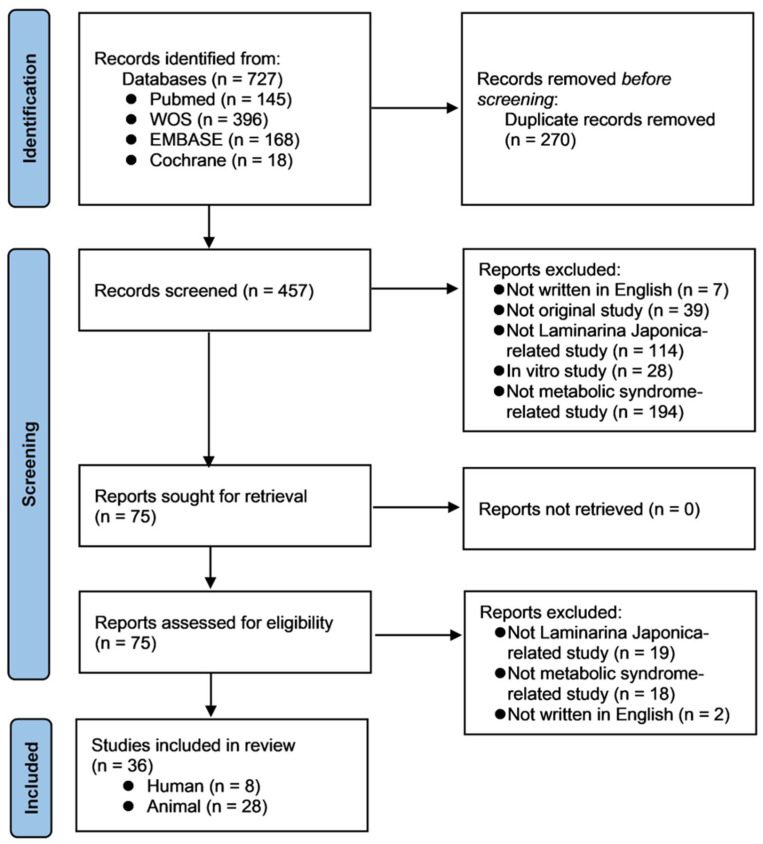
The PRISMA flow diagram of the selection and inclusion process of the studies.

**Figure 2 nutrients-14-03046-f002:**
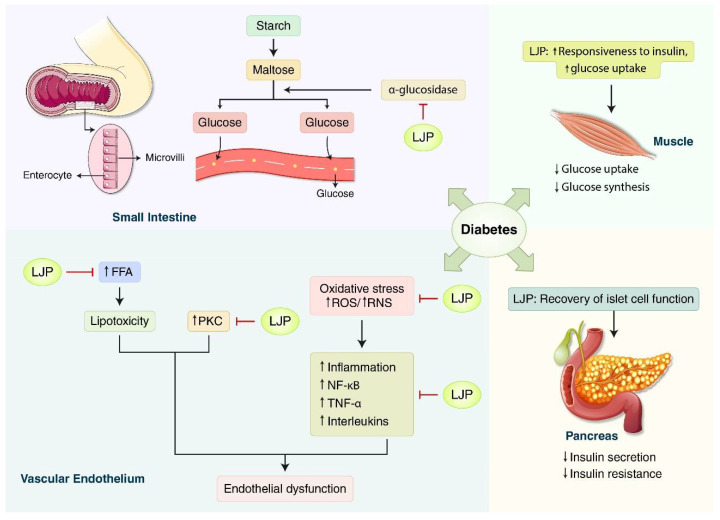
**Schematic overview of main mechanisms of *Laminaria japonica* (LJP) against diabetes.** Insulin resistance with impairment of insulin signaling, hyperinsulinemia, and hyperglycemia contributes to multiple processes including elevated free fatty acids (FFA), protein kinase C (PKC) activation, and oxidative stress, contributing to inflammation and endothelial dysfunction simultaneously. LJP may decrease the mobilization of FFA and oxidative stress, the downregulation of PKC, and the modulation of inflammatory mediators including tumor necrosis factor-α (TNF-α) and interleukins, and the nuclear factor kappa B (NF-κB) signaling pathway. Meanwhile, α-glucosidases are glycoside hydrolases found on the luminal surface of enterocytes containing maltase activities. LJP can help regulate and maintain an adequate blood sugar level by inhibiting α-glucosidase. LJP also has the potential to increase glucose uptake and responsiveness to insulin in the muscle and recover the pancreatic islet cell secreting function. FFA: free fatty acids; LJP: *Laminaria japonica*; NF-κB: nuclear factor kappa B; PKC: protein kinase C; TNF-α: tumor necrosis factor-α; ↑: increased; ↓: decreased.

**Figure 3 nutrients-14-03046-f003:**
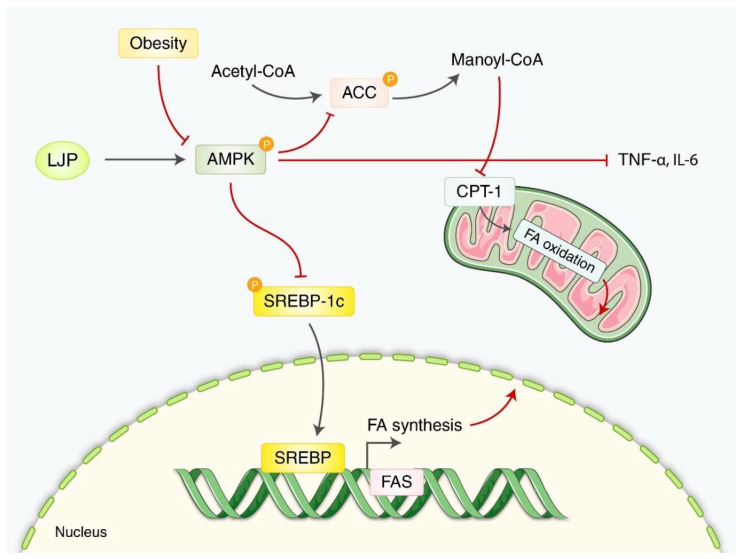
**A schematic overview of the principal mechanisms of LJP against****obesity**. In normal conditions, AMPK upregulates fatty acid oxidation by inhibiting ACC and downregulates FA synthesis by inhibiting the dephosphorylation of transcription factor SREBP-1c. Additionally, obesity inhibits the AMPK, increasing the FA synthesis and reducing FA oxidation. LJP revered the obesity-induced FA synthesis and increased the AMPK activity and FA oxidation. ACC: acetyl coenzyme A carboxylase, AMPK: AMP-activated protein kinase, CPT-1: carnitine palmitoyl transferase 1, FAS: fatty acid synthase, SREBP-1c: sterol regulatory element-binding protein 1c.

**Figure 4 nutrients-14-03046-f004:**
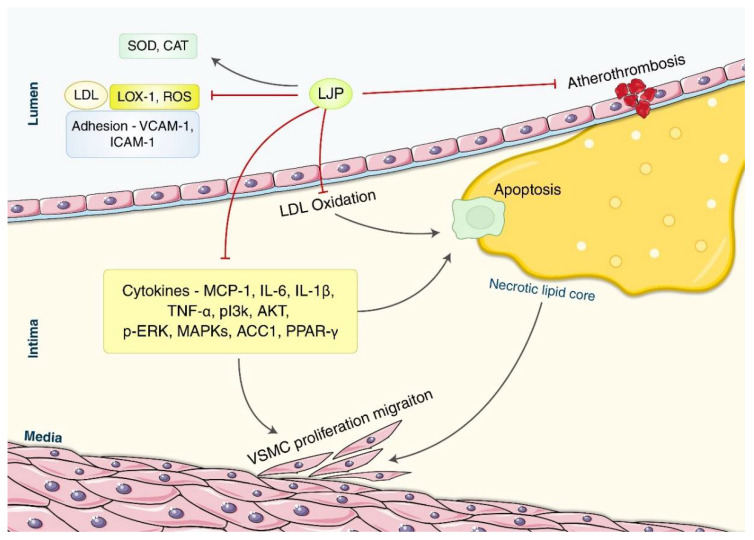
**A schematic overview of the main mechanisms of LJP against atherosclerosis**. Endothelial dysfunction causes lipid retention in the intima, the induction of endothelial expression of adhesion molecules, and the secretion of chemotactic substances, promoting leukocyte recruitment, adhesion, and transmigration into the vessel wall in patients with atherosclerosis. VSMC migrates from the media into the intima after dedifferentiating from a contractile to a proliferating phenotype. LJP inhibits atherothrombosis by interrupting oxidative stress-related lipid peroxidation and the PI3K/Akt signaling pathway, promoting cell growth, proliferation, and angiogenesis. It also regulates inflammatory and macrophage-related cytokines IL-6, IL-1β, and TNF-α. LJP treats atherosclerosis through its inhibitory action on the enhanced proliferation and migration of VSMC in atherosclerosis patients. ACC1: acetyl-CoA carboxylase; CAT: catalase; ICAM-1: intercellular adhesion molecule-1; IL: interleukin; LJP: *Laminaria japonica*; LOX-1: lectin-like oxidized low-density lipoprotein receptor-1; LPL: lipoprotein lipase; MAPKs: mitogen-activated protein kinases; MCP-1: monocyte chemoattractant proteins; (p-)ERK: (phosphorylated); PI3K/Akt: phosphoinositide 3-kinase/protein kinase B; PPAR-γ: peroxisome proliferator-activated receptors-γ; ROS: reactive oxygen species; SOD: superoxide dismutase; TNF-α: tumor necrosis factor-α; VCAM-1: vascular cell adhesion molecule-1; VSMC: vascular smooth muscle cells.

**Figure 5 nutrients-14-03046-f005:**
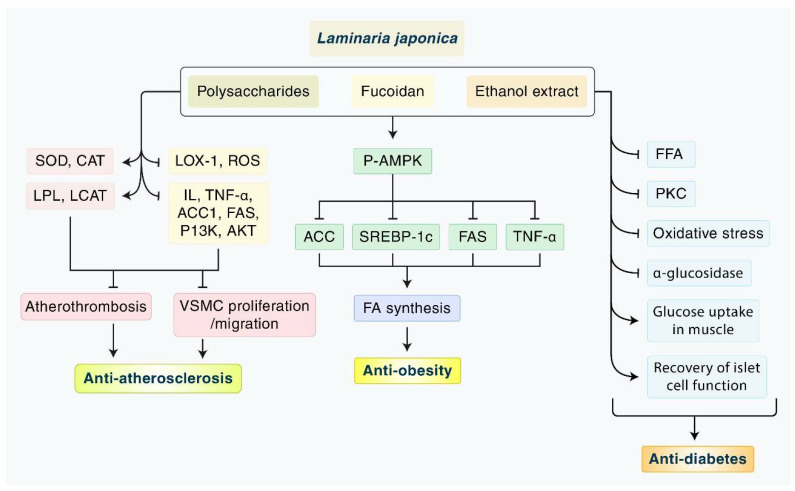
**A schematic overview of the main mechanisms of LJP against atherosclerosis, obesity, and diabetes**. LJP inhibits atherothrombosis by interrupting oxidative stress-related lipid peroxidation, the PI3K/Akt signaling pathway, inflammatory and macrophage-related cytokines, and the proliferation and migration of VSMC. LPJ exhibits the anti-obesity effect through the upregulation of P-AMPK. Subsequently, it inhibits ACC, SREBP-1c, and TNF-α, which results in decreased FA synthesis. LJP has the potential to play a role as the antidiabetic agent through the inhibition of the mobilization of FFA, PKC, oxidative stress, and α-glucosidases. LJP may increase glucose uptake and responsiveness to insulin in the muscle and recover the pancreatic islet cell secreting function, resulting in the regulation of blood sugar level and insulin resistance. ACC1: acetyl-CoA carboxylase; CAT: catalase; FFA: free fatty acids; IL: interleukin; LCAT: lecithin cholesterol acyltransferase; LJP: *Laminaria japonica*; LOX-1: lectin-like oxidized low-density lipoprotein receptor-1; LPL: lipoprotein lipase; PI3K/Akt: phosphoinositide 3-kinase/protein kinase B; PKC: protein kinase C; ROS: reactive oxygen species; SOD: superoxide dismutase; TNF-α: tumor necrosis factor-α.

**Table 1 nutrients-14-03046-t001:** An overview of the included in vivo studies on the efficacy of LJP on diabetes.

Animal Model	Diabetes-Inducer	Positive Control	Active Compounds	Administration Route	Dosage	Treatment Duration	Mechanisms	Lab Test	Efficacy	References
Kunming mice	Alloxan	Glibenclamide	Polysaccharide	Oral	50, 100, 200 mg/kg	28 days	↑ Glucose utilization, ↓ Hormone-sensitive lipase, free fatty acids mobilization	↑ Insulin, HDL-C, ↓ FBG, TC, TG, LDL-C	Hypoglycemic, hypolipidemic effect	[3]
Kunming mice	Alloxan	None	Polysaccharide	Oral	75, 150, 300 mg/mL	2 weeks	Recovery of the secretary function of islet cells	↑ Insulin, amylin, ↓ FBG	Hypoglycemic effect	[18]
Kunming mice	Streptozocin	Ethyl acetate fraction, acarbose	Butyl-isobutyl-phthalate	Intragastric	25, 50, 100 mg/kg	3 days	↓ α-glucosidase	↓ Glucose	Hypoglycemic effect	[19]
Sprague-Dawley rats	Streptozocin	None	NR	Oral	100 mg/kg	5 days	↑ Utilization efficiency of GSH, ↓ ROS	↑ GSH, GSH reductase, GSH peroxidase, XD ↓ Glucose, XO	Anti-hyperglycemic, antioxidant effect	[20]
Sprague-Dawley rats	Streptozocin	Probucol	Low molecular weight fucoidan	Intragastric	50, 100 mg/kg	12 weeks	↓ Oxidative stress, prostanoid production, hyper-responsiveness of aortic smooth muscles	↑ GSH, SOD, 6-keto-PGF_1α_, ↓ BP, TC, TG, LDL-C, MF, COX-2 expression, TXAS	Hypolipidemic, hypotensive, antioxidant effect	[21]
Sprague-Dawley rats	Streptozocin	None	NR	*Ad libitum*	4, 15% *w/w*	13 weeks	↑ Bile acid synthesis, lipid excretion, ↓ Lipid absorption	↑ Insulin, fecal TC, fecal TG, fecal TL, ↓ Glucose, TC, TG, LDL-C, Hepatic TG	Hypoglycemic, Hypolipidemic effect	[22]
Wister rats	Streptozocin	PKC inhibitor	Sulfated polysaccharide	Intragastric	200 mg/kg	80 days	Downregulation of PKC, modulation of NF-κB signaling pathway	↓ RI, Urinary volume, BUN, urinary protein/Cr, serum Cr, histopathological score, PKC-αPKC-β, NF-κB, p65, P-selectin	The effect of mitigating diabetic nephropathy	[23]
Wister rats	Alloxan	None	NR	Oral	1.25, 5.0, 12.5 g/kg	2 weeks	↑ Anti-oxidation, Recovery of the pancreatic islet cell secreting function	↑ Insulin, SOD, GSH-Px, B cell index, ↓ FBG, MDA, NO, pancreatic SOD, pancreatic iNOS	Hypoglycemic, antioxidant effect	[24]
Type 2 diabetic Goto-Kakizaki rats	Sodium laurate	Cilostazol	Low molecular weight fucoidan	Intragastric	20, 40, 80 mg/kg	4 weeks	↑ VEGF expression, eNOS phosphorylation, NO production	↑ HDL-C, NO, plantar capillary density, neovascularization around femoral artery, gastrocnemius size, weight, ↓ TG, TG, LDL-C, ulceration score, claudication score, vascular plaques rate, intimal hyperplasia thickness, ICAM-1, IL-1β, ADP	Anti-inflammation, anti-thrombosis, enhancing revascularization effect	[25]
C57BL/6N mice	High-fat diet	None	Total Polyphenol	Oral	5%	16 weeks	Regulation of α-glucose homeostasis, ↑ muscle glucose uptake, activation of insulin-signaling-related proteins	↑ IL-6, IL-10, p-Akt, p-AMPK, ↓ α-glucosidase activity, TNF-α	Antidiabetic effect	[8]

6-keto-PGF1α: a stable metabolite of prostaglandin I2; ADP: adenosine diphosphate; B cell index: the number of pancreatic islets B cells/total number of cells) × 100; BP: blood pressure; BUN: blood urea nitrogen; COX-2: cyclooxygenase-2; Cr: creatinine; eNOS: endothelial nitric oxide synthase; FBG: fasting blood glucose; FFA: free fatty acids; GSH: glutathione; GSH-Px: glutathione peroxidase; HDL-C: high-density lipoprotein-cholesterol; ICAM-1: intercellular adhesion molecule-1; IL: interleukin; iNOS: inducible nitric oxide synthase; LDL-C: low-density lipoprotein-cholesterol; LJP: *Laminaria japonica*; MDA: malondialdehyde; MF: maximum force generation of rings from endothelium-denuded thoracic aorta in response to the accumulated stimulation with phenylephrine; NF-κB: nuclear factor kappa B; NO: nitric oxide; NR: Not reported; p-Akt: phosphorylation of protein kinase B; p-AMPK: phosphorylation of 5′ adenosine monophosphate-activated protein kinase; PKC: protein kinase C; RI: renal index, renal weight × 100/body weight; ROS: reactive oxygen species; SOD: superoxide dismutase; TC: total cholesterol; TG: triglycerides; TL: total lipid; TNF-α: tumor necrosis factor- α; TXAS: thromboxane synthase; VEGF: vascular endothelial growth factor; XD: xanthine dehydrogenase; XO: xanthine oxidase. ↑: increased; ↓: decreased.

**Table 2 nutrients-14-03046-t002:** An overview of the included in vivo studies on the efficacy of LJP on obesity.

Animal Model	Obesity-Inducer	Active Compounds	Administration Route	Dosage	Treatment Duration	Positive Control	Mechanisms	Lab Test	Efficacy	Reference
C57BL/6N mice	HFD		Oral	supplementing 5% of the diet	16 weeks	None	↓ IL-1β, Il-6	↓ blood glucose, leptin	Anti-obesity effect Anti-inflammatory effect	[13]
C57BL/6J mice	HFD	Fucoidan	Oral	200 mg/kg	16 weeks	None	↓ TNF-α, IL-1β, MCP-1	↓ TC, TAG, fasting blood glucose, serum LBP	Beneficial effect on MetS	[26]
SD rats	HFD	Ethanol extract	Oral	400 mg/kg	6 weeks	None	↑ p-AMPK/AMPK, p-ACC/ACC ↓ TNF-α	↓ serum TG, TC, LDL-C, FFA, leptin, glucose, insulin ↑ HDL-C and HDL-C/TC ratio, adiponectin	Anti-obesity effect	[27]
C57BL/6 mice	HFD	Insoluble dietary fiber	Oral	supplementing 5% of the diet	8 weeks	None	regulation of SREBP-1c/FAS signaling	↓ serum glucose, TC, HDL-C, LDL-C, ALT, AST ↑ acetate, propionate, cecal SCFA	Anti-obesity effect Gut microbiota dysbiosis	[28]
C57BL/6J mice	HFD	Polysaccharide	Oral	HFD plus 2 g/kg SP	8 weeks	None	↑ p-AMPK ↓ FAS, TNF-α	↑ adiponectin secretion ↓ TG, TC, FFA, leptin secretion, Hepatic TG, cholesterol content in the liver, serum LDL-C	Hypoglycemic effect, improved serum lipid profiles, ameliorated intestinal damage	[29]
BALB/c mice	high-fat diet	Polysaccharide	Oral	0.25% LJPs solution as drinking water	10 weeks (not specified)	None		↑ ratio of HDL-C/LDL-C, SCFAs ↓ levels of serum lipids,	Gut microbiota normalization Anti-obesity effect	[30]
C57BL/6N mice	High-fat diet	N/A	Oral	supplementing 5% of the diet	16 weeks	None	↑ p-AMPK	↑ Fecal BA ↓ Fasting blood glucose levels, Plasma TG levels, hepatic lipid accumulation	Hypotriglyceridemic effect	[31]

ACC: acetyl-CoA carboxylase; ALT: alanine aminotransferase; AMPK: AMP-activated protein kinase; AST: aspartate aminotransferase; FAS: fatty acid synthase; FFA: free fatty acid; HDL-C: high-density lipoprotein cholesterol; HFD: high-fat diet; LBP: lipopolysaccharide-binding protein; LDL-C: low-density lipoprotein-cholesterol; MetS: diet-induced metabolic syndrome; N/A: not available; SCFA: short-chain fatty acid; SP: seaweed low-molecular-weight polysaccharide; SREBP: sterol regulatory element-binding protein-1; TAG: triacylglycerol; TC: total cholesterol; TG: triglyceride; ↑: increased; ↓: decreased.

**Table 3 nutrients-14-03046-t003:** An overview of the included in vivo studies on the efficacy of LJP on atherosclerosis.

Animal Model	Atherosclerosis Inducer	Active Compounds	Administration Route	Dosage	Treatment Duration	Positive Control	Mechanisms	Lab Test	Efficacy	Reference
Guangdong mice	ROS/RNS	Polysaccharide	Oral	200 mg/kg/body mass/day	4 weeks	None	↑ ABTS	↓ TC, HDL-C, TG, LDL-C/HDL-C ratio	Anti-cardiovascular diseases, hypolipidemic, antioxidative effects	[14]
C57BL/6 mice	HFD	Polysaccharide	Oral	200 mg/kg/body mass/day	8 weeks	None	↑ intestinal goblet cells ↓ ACC1, FAS, PPAR-γ, TNF-α, MAPKs, p-ERK, p-JNK, Akkermansia	↓ glycemia, glucose, fasting insulin/glucose, HOMA-IR, inflammation, Firmicutes/Bacteroidetes ratio	Anti-insulin resistance, anti-obesity, anti-inflammation effects	[32]
LDLr−/− mice	HFD	Polysaccharide	Oral	200 mg/kg/body mass/day	14 weeks	None	↑ SOD ↓ MAPKs, TNF-α, p- ERK1/2, p-JNK1/2	↓ atherosclerotic plaque, TC, TG, LDL-C/HDL-C, MDA	Anti-atherosclerotic, hypolipidemic, antioxidative effects	[33]
Sprague-Dawley rats	HFD	FPS	Oral	0.4 g/kg	8 weeks	None	↑ LPL, LCAT	↑ HDL-C ↓ TG, TC, LDL-C, synthesis of endogenous lipids	Hypoglycemic, anti-atherosclerotic cardiovascular diseases effects	[34]
BALB/c mice	HFD	L-LJA	Oral	0.3%	11 weeks	None	↑ GPR41, GPR43, CPT-1A ↓ PPAR-γ, TNF-α	↑ HDL-C, SOD, CAT, SCFAs ↓ TC, TG, LDL-C	Anti-obesity effect	[35]
Kunming mice	Hyperlipidemic diets	Polysaccharides	Oral	100, 200, 400 mg/kg/day	12 weeks	None	↑ SOD, CAT, GST	↓ TC, TG, HDL-C, LDL-C, MDA	hypolipidemic, enhancing antioxidant enzyme effects	[36]
LDL receptor-deficient C57BL6J mice	HCD	Polysaccharide (Fucoidan)	Oral	50, 100 mg/kg/day	16 weeks	None	↓ LOX-1, IL-1b, IL-6, TNF-α, ICAM-1, VCAM-1	↓ TG, TC, LDL-C, HDL-C, atherosclerotic plaque formation, macrophage infiltration, smooth muscle cell proliferation, ROS generation	Anti-atherosclerotic, hypolipidemic, anti-inflammatory effects	[37]
LDL receptor-deficient C57BL6J mice	HCD	Polysaccharide (Fucoidan)	Oral	50, 100 mg/kg/day	14 weeks	Simvastatin (5 mg/kg/day)	↓ VLDL, SREBP-1c, ACC1, FAS, p-IRS-1, p-IRS-2, PI3K, AKT, P70S6K, nuclear Foxo1	↑ Apolipoprotein A1, HDL, Sortilin ↓ insulin resistance, fat accumulation, plaque, HDL-C, FFA, hepatic cholesterol/TC, VLDL-CE/FC/TG/apolipoprotein B	Anti-atherosclerotic, hypolipidemic, insulin signaling regulating effects	[38]

ABTS: 2-2′-azinobis-(3-ethylbenzthiazo- line-6-sulfonate); ACC1: Acetyl-CoA carboxylase; AKT: Protein kinase B; CAT: catalase; CE: cholesteryl ester; CPT-1A: carnitine palmitoyltransferase-1A; FAS: fatty acid synthase; FC: free cholesterol; FFA: free fatty acid; Foxo1: forkhead box protein O1; GPR: G protein-coupled receptor; GST: glutathione S transferase; HCD: high-cholesterol diet; HDL-C: high-density lipoprotein-cholesterol; HFD: High-fat diet; HOMA-IR: homeostasis model assessment of insulin resistance; ICAM-1: intercellular adhesion molecule-1; IL: interleukin; IRS: Insulin receptor substrate; JNK: c-Jun N-terminal kinase; LCAT: lecithin cholesterol acyltransferase; LDL-C: low-density lipoprotein-cholesterol; L-LJA: low- molecular alginate from *Laminaria japonica*; LOX-1: lectin-like oxidized low-density lipoprotein receptor-1; LPL: lipoprotein lipase; LPS: lipopolysaccharide; MAPKs: mitogen- activated protein kinases; MDA: malonaldehyde; ORAC: oxygen radical absorbance capacity; (p-)ERK: (phosphorylated) extracellular signal-regulated kinase; PI3K: phosphoinositide 3-kinase; PPAR-γ: peroxisome proliferator-activated receptors-γ; RNS: reactive nitrogen species; ROS: reactive oxygen species; SCFA: short chain fatty acids; SD: Sprague-Dawley; SOD: superoxide dismutase; SREBP-1c: sterol regulatory element-binding protein 1; TC: total cholesterol; TG: triglycerides; TNF-α: tumor necrosis factor α; VCAM-1: vascular cell adhesion molecule-1; VLDL: very low density lipoprotein; ↑: increased; ↓: decreased.

**Table 4 nutrients-14-03046-t004:** An overview of the included in vivo studies on the efficacy of LJP on hyperlipidemia.

Animal Model	Obesity-Inducer	Active Compounds	Administration Route	Dosage	Treatment Duration	Positive Control	Mechanisms	Lab Test	Efficacy	Reference
SD rats	HFD	N/A	Oral	2.5 g/kg	8 weeks	None	↑ SOD, GSH-Px	↓ TG, TC, NEFA	Hypolipidemic effect	[39]
SPF male rats	HFD	N/A	Oral	1.0 mL	8 weeks	None	↓ HMGCR, SREBP-1c, CD36	↓ serum TC, TG, NEFA ↑ fecal acid acetate, propionate, isobutyrate	Anti-hyperlipidemia effect	[40]
ICR mice	Non-alcoholic fatty liver/high fat-diet	None	Oral	50 mg/kg	4 weeks	None	↑ AMPK and regulation of its downstream proteins	↑ p-AMPK, PPAR-α, APT-1, ↓ body weight, liver index, visceral fat index, plasma, and hepatic TC, TG, HDL, LDL, hepatic steatosis, accumulation of hepatic lipids, hepatocellular swelling, vacuoles (normal diet + LJP group)	Hypolipidemic effect	[41]

GSH-Px: glutathione peroxidase; NEFA: non-esterified fatty acids; SD: Sprague-Dawley; SOD: superoxide dismutase; TC: total cholesterol; TG: triglycerides; ↑: increased; ↓: decreased.

**Table 5 nutrients-14-03046-t005:** An overview of the included clinical trials on the efficacy of LJP on metabolic syndrome.

Patient /Inclusion Criteria	Intervention (n)	Control (n)	Treatment Period	Outcome	Main Results	Adverse Effect	Reference/Research Design
Healthy, female (*n* = 22) /NR	Sea tangle (20 g/day)	-	8 weeks	Body composition, dietary intakes, QOL	↓ body weight/fat, BMI, intake of energy/protein/fat ↑ balanced diet/mealtime, intake of fiber improvement: QOL	NR	[42]/CT
Healthy, high GGT/aged 25–60 year	Fermented sea tangle (250 mg×6)	Placebo	4 weeks	Oxidative stress	↓ GGT, MDA ↑ SOD, CAT activities	No	[47]/RCT
Healthy (*n* = 40) /aged 18–75 year	LJP (625 mg)	LJP + probiotics (lactic acid bacteria)	4 weeks (+2 weeks follow-up)	Gastrointestinal symptom, QOL, microbiome	No significant changes	No	[48] /RCT
Healthy, female (*n* = 21) /middle aged	γ-aminobutyric acid-enriched fermented sea tangle (1000 mg/day)	Placebo (sucrose)	8 weeks	Body composition, muscular strength	↓ fat, TG ↑ BDNF, ACE, HGH, IGF-1, total lean mass improvement: total work, muscle strength	No	[50] /RCT
Healthy (*n* = 70) /LDL-C 120~160 mg/dL, BMI 22~30	Dried kombu powder (2.0 g/day)	Placebo (dextrin powder)	6 weeks	Liver/renal function, body composition, lipid/glucose profiles	↓ fat ↑ adiponectin	diarrhea, variation in LDH/γ-GTP/UA	[43] /RCT
Healthy (*n* = 48) /NR	Roasted kombu (6 g/day)	-	4 weeks	Liver/renal function, lipid/glucose profiles, insulin, gastrointestinal symptoms	↓ UA ↓ serum triglyceride (only in subjects with high serum triglyceride levels) ↑ TC, CPR, abdominal distension	No	[44] /RCT, cross-over
1) Healthy (*n* = 48) 2) high TG (*n* = 9)	Roasted kombu (6 g/day, first 4 weeks)	Roasted kombu (6 g/day, last 4 weeks)	4 weeks	Lipid metabolomics	↑ Plasmanyl/plasmenyl forms of PC, PE ↓ LPC, LPE improvement: PC/PE with DL, LPC/LPE with AL, FFA (high TG subjects)	NR	[45] /RCT, cross-over (re-analysis)
Healthy (n = 50) /BMI < 30 kg/m^2^	iodine-reduced boiled kelp powder (3 g alginate/day)	Placebo	8 weeks	Lipids, thyroid hormone	↓ body fat (male subjects)	No	[46] /RCT

ACE: angiotensin converting enzyme; AL: acyl linkages; BDNF: brain-derived neurotrophic factor; BMI: body mass index; CAT: catalase; CPR: C-peptide immunoreactivity; CT: controlled clinical trials; DL: diacyl linkages; FFA: free fatty acid; GGT: gamma-glutamyl transferase/transpeptidase; GPx: glutathione peroxidase; GTP: g-glutamyl transpeptidase; HDL-C: high-density lipoprotein cholesterol; HGH: human growth hormone; Ht: hematocrit; IGF: insulin-like growth factor; IL: interleukin; LDH: lactate dehydrogenase; LDL-C: low-density lipoprotein cholesterol; LJP: *Laminaria japonica*; LPC: lysophosphatidylcholines; LPE: lysophosphatidylethanolamine; MCH: mean corpuscular hemoglobin; MCHC: mean corpuscular hemoglobin concentration; MCV: mean corpuscular volume; MDA: malondialdehyde; NR: not reported; PC: phosphatidylcholines; PE: phosphatidylethanolamines; QOL: quality of life; RCT: randomized controlled clinical trials; RBC: red blood cell; SOD: superoxide dismutase; TC: total cholesterol; TG: triglyceride; WBC: white blood cell; UA: uric acid; y: year(s); γ-GTP: γ-glutamyl transpeptidase; 1,5-AG: 1,5-anhydroglucitol.

**Table 6 nutrients-14-03046-t006:** The risk of bias assessment for the randomized clinical trials using the ROB 2 criteria.

Author, Year/Research Design	Randomization Process	Deviations from Intended Interventions	Missing Outcome Data	Measurement of the Outcome	Selection of the Reported Result	Period & Carryover Effects	Overall
[47]/RCT	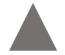	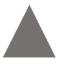	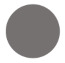	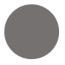	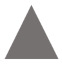	-	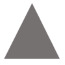
[48]/RCT	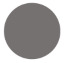	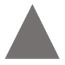	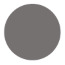	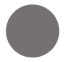	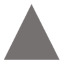	-	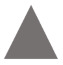
[50] /RCT	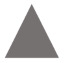	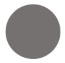	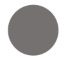	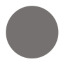	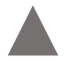	-	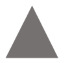
[43]/RCT	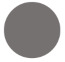	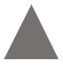	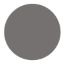	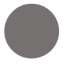	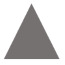	-	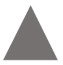
[46]/RCT	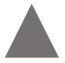	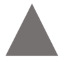	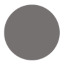	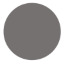	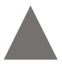	-	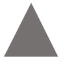
* [44]/RCT, cross-over	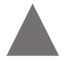	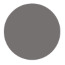	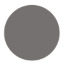	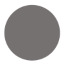	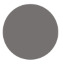	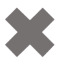	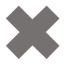

* RoB 2 for cross-over RCT was used for this study. Cross symbols: high risk of bias; triangle symbols: some concerns; circle symbols: low risk of bias.

## Data Availability

Not available.

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
