# Peer review of "The Effect of Laminaria japonica on Metabolic Syndrome: A Systematic Review of Its Efficacy and Mechanism of Action"

_nutrients, 2022, doi:10.3390/nu14153046_

Round 1
Reviewer 1 Report
This is a review study discussing about the effects and mechanisms of LJP in the treatment of patients with metabolic syndrome, and the possibility of using LJP as a therapeutic drug. I think the topic is interesting and contributive to etiology of adults’ Met status with a clinical study carried mechanism articles.
Minor comments
I suggest adding treatment solution for present challenges and future perspectives for adults’ Met prognosis for nutrients extracted from Laminaria Japonica.
Table 3 and 4 seems truncated margins. Please revise accordingly.
Author Response
Dear Reviewer
We appreciate reviewers and editors for giving us an opportunity to resubmit our manuscript, entitled “The effect of Laminaria Japonica on metabolic syndrome: A systematic review of its efficacy and mechanism of action”. We earnestly responded to the raised comments point by point.
Minor comments
- I suggest adding treatment solution for present challenges and future perspectives for adults’ Met prognosis for nutrients extracted from Laminaria Japonica.
(Response): Following your instruction, we added treatment solutions for Met-related diseases for nutritional compounds extracted from Laminaria japonica in the discussion section as follows.
[Before] – Discussion section
… They found that boiled LJP had a higher bioavailability than steamed and soaked LJP. However, further studies are required [83].
Although the interaction of LJP with other drugs or herbs has rarely been studied, …
[After] – Discussion section
… They found that boiled LJP had a higher bioavailability than steamed and soaked LJP. However, further studies are required [83]. In addition to fucoidan and iodine, alginates, polyphenols and polysaccharides as bioactive nutritional compounds extracted from LJP have been evaluated as solutions for the treatment and prevention of Met-related diseases [84]. In this review, alginates, polyphenols and polysaccharides have been reported to possess anti-diabetic [3,10,20,25,35], anti-obesity [30–34], antioxidant [16,36,39], anti-inflammatory [34,40] and hypolipidemic [16,35,36,39,40] effect.
Although the interaction of LJP with other drugs or herbs has rarely been studied, …
[Newly inserted reference]
- Wijesinghe, W.A.J.P.; Jeon, Y.J. Biological activities and potential cosmeceutical applications of bioactive components from brown seaweeds: A review. Phytochem. Rev. 2011, 10, 431–443. doi: 10.1007/s11101-011-9214-4.
- Table 3 and 4 seem truncated margins. Please revise accordingly.
(Response): Following your instruction, we have added a bold line below the table 2–4 to avoid truncated margins.
Again, we appreciate reviewers and editors for their kind and careful comments for improving the quality of our manuscript and also sincerely hope we address our responses well to the raised comments and our revised manuscript would be accepted for publication in your journal soon.
With kind regards,
Prof. Bonglee Kim, M.D, Ph.D.
-Associate Professor of Department of Pathology, College of Korean Medicine, Kyung Hee University, 26 Kyungheedae-ro, Dongdaemun-gu, Seoul, 02453, Republic of Korea
-Chair of Department of Cancer Preventive Material Development, Kyung Hee University
-Group leader of Korean Medicine-Based Drug Repositioning Cancer Research Center
Phone: +82-2-961-9217 (South Korea)
E-mail: bongleekim@khu.ac.kr
Reviewer 2 Report
1-Please explain more about rational of the study topic tips of this study is not very clear please explain more
2-please explain the main protocol of the study. The main points should be Elaborated
3-It would be better if you don’t use the words such as in the metal part and write down exactly which keywords you used
4-table 1-5 seems to be very large can you summarize the information? Maybe you can bring some information to the supplementary material.
5-please explain more about how you made table 6
Author Response
Dear Reviewer
We appreciate reviewers and editors for giving us an opportunity to resubmit our manuscript, entitled “The effect of Laminaria Japonica on metabolic syndrome: A systematic review of its efficacy and mechanism of action”. We earnestly responded to the raised comments point by point.
1-Please explain more about rational of the study topic tips of this study is not very clear please explain more
(Response): Thank you for your valuable comments. We added rational of this study topic tips clearly in the introduction section.
“LJP has multiple pharmacological effects, including anti-tumor, anti-thrombotic, anti-atherosclerosis, hypolipidemic, hypoglycemic, antioxidant, anti-inflammatory, renoprotective, and immunomodulatory [14]. In addition, LJP protects against several metabolic syndromes, including diabetes, obesity, and atherosclerosis [3,15,16]. Some in vitro studies also identified an antithrombotic effect in LJP [17]. However, there is no systematic review of studies testing for the effects of LJP on MetS, and we aimed to address the role of LJP on the treatment of MetS.
In this study, we conducted a systematic review to determine the effects and mechanisms of LJP in treating patients with MetS and the possibility of using LJP as a therapeutic drug. We also discuss the mechanisms to delve deeper into its molecular pharmacology and highlight the recent progression of its therapeutic developments. This review is expected to direct future research toward a better understanding of how LJP could modulate MetS and be developed as a potential therapeutic component for the treatment of MetS.”
2-please explain the main protocol of the study. The main points should be Elaborated
(Response): We appreciate the reviewer's insightful criticism. The entire manuscript was revised in response to the reviewer's comment, and we are hopeful that the revised manuscript better presents the main points of the study.
(Changes in the abstract are marked in the manuscript)
Page 5
However, there is no systematic review of studies testing for the effects of LJP on MetS, and we aimed to address the role of LJP on the treatment of MetS.
A systematic review was conducted to retrieve and critically appraise the available evidence on the effects of LJP on MetS.
Page 6
To ensure that the results of evidence synthesis are transparent, risk of bias assessment was performed for randomized controlled trials (RCT) using the Revised Cochrane risk-of-bias tool for randomized trials (RoB 2) for individually randomized RCTs (Sterne et al., 2019).
Page 37
MetS include hypertension, abdominal obesity, hyperlipidemia, and insulin resistance (Saklayen, 2018). Impaired metabolic homeostasis in the human body is the leading cause of this syndrome (Huang et al., 2021a;Huang et al., 2021b). In this systematic review, we found that LJP could significantly improve clinical outcomes in patients suffering from MetS and laboratory measurements in vivo models of MetS.
3-It would be better if you don’t use the words such as in the metal part and write down exactly which keywords you used
(Response): Sorry for the mistake. “Such as” are modified and keywords are exactly used.
4-table 1-5 seems to be very large can you summarize the information? Maybe you can bring some information to the supplementary material.
(Response): We thank the reviewer for pointing this out. We politely request that the current tables be maintained as-is, with the exception of Table 5, since the tables represent the study's primary findings. We shortened the Table 5, especially the ‘Outcome’ and ‘Main results’ columns, and we hope that the reviewer finds the changes to improve the readability of the tables.
(Changes are marked in the Table 5 in the manuscript)
5-please explain more about how you made table 6
(Response): We thank for the reviewer for this suggestion. We added more details about how the table 6 was made in the Methods.
Quality Assessment and Risk of Bias
“To ensure that the results of evidence synthesis are transparent, risk of bias assessment was performed for randomized controlled trials (RCT) using the Revised Cochrane risk-of-bias tool for randomized trials (RoB 2) for individually randomized RCTs (Sterne et al., 2019). RoB 2 for cross-over trials was used in the cross-over RCT study. Following the guidelines, we assessed each study's risk of bias based on the randomization process, deviations from intended interventions, missing outcome data, outcome measurement, selection of the reported result, and period and carry-over effects (only for the cross-over study). Cochrane Methods (https://methods.cochrane.org/bias/resources/rob-2-revised-cochrane-risk-bias-tool-randomized-trials) offered an Excel tool that we used to calculate the risk of bias and to generate a table of the assessment result (Table 6).”
Again we appreciate reviewers and editors for their kind and careful comments for improving the quality of our manuscript and also sincerely hope we address our responses well to the raised comments and our revised manuscript would be accepted for publication in your journal soon.
With kind regards,
Prof. Bonglee Kim, M.D, Ph.D.
-Associate Professor of Department of Pathology, College of Korean Medicine, Kyung Hee University, 26 Kyungheedae-ro, Dongdaemun-gu, Seoul, 02453, Republic of Korea
-Chair of Department of Cancer Preventive Material Development, Kyung Hee University
-Group leader of Korean Medicine-Based Drug Repositioning Cancer Research Center
Phone: +82-2-961-9217 (South Korea)
E-mail: bongleekim@khu.ac.kr
Huang, C., Chen, X., Wei, C., Wang, H., and Gao, H. (2021a). Deep Eutectic Solvents as Active Pharmaceutical Ingredient Delivery Systems in the Treatment of Metabolic Related Diseases. Frontiers in pharmacology 12.
Huang, C., Chen, X., Wei, C., Wang, H., and Gao, H.J.F.I.P. (2021b). Deep Eutectic Solvents as Active Pharmaceutical Ingredient Delivery Systems in the Treatment of Metabolic Related Diseases. 12, 794939-794939.
Saklayen, M.G.J.C.H.R. (2018). The global epidemic of the metabolic syndrome. 20, 1-8.
Sterne, J.A., Savović, J., Page, M.J., Elbers, R.G., Blencowe, N.S., Boutron, I., Cates, C.J., Cheng, H.-Y., Corbett, M.S., and Eldridge, S.M.J.B. (2019). RoB 2: a revised tool for assessing risk of bias in randomised trials. 366.
Sun, N., Sun, B., Li, C., Zhang, J., and Yang, W.J.J.O.a.F.P.T. (2022). Effects of Different Pretreatment Methods and Dietary Factors on the Form and Bioavailability of Iodine in Laminaria japonica. 1-16.
Round 2
Reviewer 2 Report
Dear authors,
Tank you for providing the revised version,
My points have been all considered